# FMBench: Benchmarking Fairness in Multimodal Large Language Models on Medical Tasks

## Abstract

Advancements in Multimodal Large Language Models (MLLMs) have significantly improved medical task performance, such as Visual Question Answering (VQA) and Report Generation (RG). However, the fairness of these models across diverse demographic groups remains underexplored, despite its importance in healthcare. This oversight is partly due to the lack of demographic diversity in existing medical multimodal datasets, which complicates the evaluation of fairness. In response, we propose **FMBench**, the first benchmark designed to evaluate the fairness of MLLMs performance across diverse demographic attributes. FMBench has the following key features: **(1):** It includes four demographic attributes: race, ethnicity, language, and gender, across two tasks, VQA and RG, under zero-shot settings. **(2):** Our VQA task is free-form, enhancing real-world applicability and mitigating the biases associated with predefined choices. **(3):** We utilize both lexical metrics and LLM-based metrics, aligned with clinical evaluations, to assess models not only for linguistic accuracy but also from a clinical perspective. Furthermore, we introduce a new metric, **Fairness-Aware Performance (FAP)**, to evaluate how fairly MLLMs perform across various demographic attributes. We thoroughly evaluate the performance and fairness of eight state-of-the-art open-source MLLMs, including both general and medical MLLMs, ranging from 7B to 26B parameters on the proposed benchmark. We aim for **FMBench** to assist the research community in refining model evaluation and driving future advancements in the field. All data and code will be released upon acceptance.

## 1 Introduction

Significant progress has been made in Multimodal Large Language Model (MLLM) (Wang et al., 2024; Yao et al., 2024), exemplified by models such as the InternVL series and Mini-CPM (Yao et al., 2024; Chen et al., 2024). These advancements in general MLLMs have also spurred developments in the medical domain, as seen with LLaVA-Med (Li et al., 2024). Although general and medical MLLMs are commonly assessed on vision-language tasks like Visual Question Answering (VQA) and report generation (RG), their fairness across diverse demographic groups has been less explored (Hu et al., 2024), despite its critical importance in clinical applications (Cheng et al., 2024). Previous studies on fairness in the medical field have predominantly focused on classical single-modality tasks and have not sufficiently addressed multimodal tasks (Chen et al., 2023). Given that MLLMs are trained on large-scale and diverse datasets, a pertinent question arises: ***Do MLLMs perform fairly on medical multimodal tasks?***

To the best of our knowledge, there is currently no public benchmark for evaluating fairness comprehensively in medical multimodal tasks, which include various demographic attributes. To address this gap, our main contributions are:

- We introduce **FMBench**, the first benchmark specifically designed to evaluate the fairness of MLLMs on medical multimodal tasks, including VQA and RG. FMBench comprises a dataset of 30,000 medical VQA pairs and 10,000 medical image-report pairs, each annotated with detailed demographic attributes (race, gender, language, and ethnicity) to facilitate a thorough evaluation of MLLM fairness.

- We propose **Fairness-Aware Performance (FAP)**, a novel metric designed to assess the equitable performance of MLLMs across different demographic groups, filling the gap left by the lack of existing metrics to evaluate MLLM fairness on open-form multimodal tasks.

- We benchmark eight mainstream MLLMs, ranging from 7B to 26B parameters, including both general-purpose and medical models. These models are evaluated using traditional lexical metrics, clinician-verified LLM-based metrics, and our proposed FAP metric. Experimental results reveal that traditional lexical metrics are insufficient for open-form multimodal tasks and may even conflict with clinician-verified metrics. Furthermore, all MLLMs exhibit inconsistent performance across different demographic attributes, indicating potential fairness risks.

## 2 RELATED WORK

**Medical Visual Question Answering.** Medical Visual Question Answering (MedVQA) involves answering questions based on medical images and associated queries (Zhang et al., 2023). Recent developments include datasets such as VQA-RAD (Lau et al., 2018), Path-VQA (He et al., 2020), SLAKE (Liu et al., 2021a), and OmniMedVQA (Hu et al., 2024). These datasets, however, lack demographic information, complicating the evaluation of model fairness across different population groups. Additionally, they predominantly feature closed-form answers, which contrasts with the open-ended responses required in real clinical scenarios (Oh et al., 2024). The absence of demographic data and reliance on closed-form questions underscore the need for a dataset capable of assessing real-world performance and fairness in medical VQA tasks.

**Medical Report Generation.** Much of the research in medical report generation (RG) has concentrated on radiology, particularly chest X-ray report generation (Liu et al., 2021b; Chen et al., 2020; Boecking et al., 2022). These studies typically do not assess fairness across diverse population groups, thus limiting their generalizability and real-world applicability (Seyyed-Kalantari et al., 2020; Badgeley et al., 2019). This limitation largely stems from the lack of comprehensive demographic data in major RG datasets, which hampers fairness assessments (Huang et al., 2021; Irvin et al., 2019). Addressing this, our work introduces FMBench, the first benchmark to comprehensively evaluate fairness in medical report generation tasks.

**Multimodal Large Language Models (MLLMs).** MLLMs have shown significant advancements in vision-language tasks (OpenAI, 2023), with models like LLaVA (Liu et al., 2024), InternVL (Chen et al., 2024), and MiniCPM-V (Yao et al., 2024), including medical-specific versions such as LLaVA-Med (Li et al., 2024). Despite being trained on diverse, web-scale datasets, these models still encounter issues with unfairness and social biases across different demographic groups (Cheng et al., 2024). Given the critical role of fairness in healthcare, where biased predictions can result in detrimental outcomes, our work presents the first benchmark specifically designed to assess the fairness of MLLMs in medical multimodal tasks.

## 3 FMBENCH

In this section, we describe the benchmark construction pipeline and provide detailed information, including the creation of VQA pairs and the introduction of our new FAP metric to evaluate the fairness of MLLMs on open-form multimodal tasks. We developed a series of high-quality question-and-answer pairs using the open-source fundus medical visual language dataset known as the Harvard-FairVLMed dataset (Luo et al., 2024), from the Massachusetts Eye and Ear Infirmary at Harvard Medical School.

### 3.1 DATA SOURCE

FMBench is constructed using the Harvard-FairVLMed dataset (Luo et al., 2024), which comprises 10,000 samples. Each sample includes a fundus image paired with a clinical report, supplemented by metadata such as race, gender, ethnicity, and language. As indicated in Table 1, there are few medical multimodal datasets that encompass multiple demographic attributes. FMBench represents the first initiative to integrate such diverse data into a dataset specifically designed for Multimodal Large Language Model applications. Additionally, two representative samples from the dataset are

illustrated in Figure 1 (a). All original data is publicly accessible[1]. The demographic data for each sample is meticulously detailed, with each attribute segmented into multiple groups:

**Race:** White, Asian, and Black.

**Gender:** Male and Female.

**Ethnicity:** Non-Hispanic and Hispanic.

**Language:** English, Spanish, and Other.

With the detailed demographic data, we aim to benchmark the performance of various MLLMs on two tasks: VQA and RG, across different demographic groups to evaluate the fairness of MLLMs.

| Benchmarks | Images | QA pairs | Demographic |
|---|---|---|---|
| VQA-RAD (Lau et al., 2018) | 0.3k | 3.5k | - |
| Path-VQA (He et al., 2020) | 5k | 32k | - |
| SLAKE (Liu et al., 2021a) | 0.6k | 1.4k | - |
| OmniMedVQA (Hu et al., 2024) | 118k | 128k | - |
| **FMBench (ours)** | 10k | 30k | Race, Gender, Ethnicity, Language |

Table 1: Overview of current available datasets to evaluate MLLM capabilities on medical multimodal tasks. The table lists the number of images and QA pairs for each dataset. Unlike others, FMBench includes demographic data (Race, Gender, Ethnicity, Language) to assess MLLM fairness. '-' indicates that those datasets do not provide demographic data.

### 3.2 QA PAIR GENERATION AND OPTIMIZATION

**Constructing QA Pairs.** To construct QA pairs based on clinical reports, we follow the method outlined by (Oh et al., 2024), querying an LLM with existing clinical reports to generate QA pairs. Specifically, we employ Llama-3.1-Instruct-70B (Meta, 2024) as the LLM for generating these pairs. We prompt the LLM with the following instruction: `You're a helpful AI Ophthalmologist. Please generate 3 concise Question and Answer pairs based on the given clniical reports. The questions must belong the following three categories and each category only appear one time: 1. Primary Condition or Diagnosis, 2. Testing or Treatment, 3. Medical Condition. The given clinical report is <Clinical Report>.` We illustrate the construction process and show three example QA pairs in Figure 1 (b).

**Post-processing.** To enhance the quality of the generated open form QA pairs, we instruct Llama3.1-70B-Instruct to perform a self-check of its initial output of these QA pairs in conjunction with the report. Overall, our benchmark includes 10k image-report pairs, and 30k VQA pairs with 3 types, 4 different demogrphy attributes. This allowed us to comprehensively assess the fairness of MLLM performance on two mulitmodal tasks, VQA and report generation.

### 3.3 FAIRNESS-AWARE PERFORMANCE

To evaluate the fairness of Multimodal Large Language Models (MLLMs) across various demographic groups in Visual Question Answering (VQA) and Report Generation (RG) tasks, traditional metrics such as BLEU and METEOR (Papineni et al., 2002; Banerjee & Lavie, 2005) prove insufficient as they primarily assess linguistic correctness rather than fairness. Moreover, merely averaging performance across different demographic groups can obscure significant disparities. To address this, we introduce the Fairness-Aware Performance (FAP) metric, designed to quantitatively assess the fairness of MLLM performance.

To compute FAP, we first calculate the performance scores for each individual group $\mathcal{G}_i$, which reflect the effectiveness of MLLMs on specified tasks. In this study, we utilize the GREEN score (Ostmeier et al., 2024) to evaluate each group's performance. These scores are weighted ($\mathcal{W}_i$)

---

[1]https://github.com/Harvard-Ophthalmology-AI-Lab/FairCLIP?tab=readme-ov-file

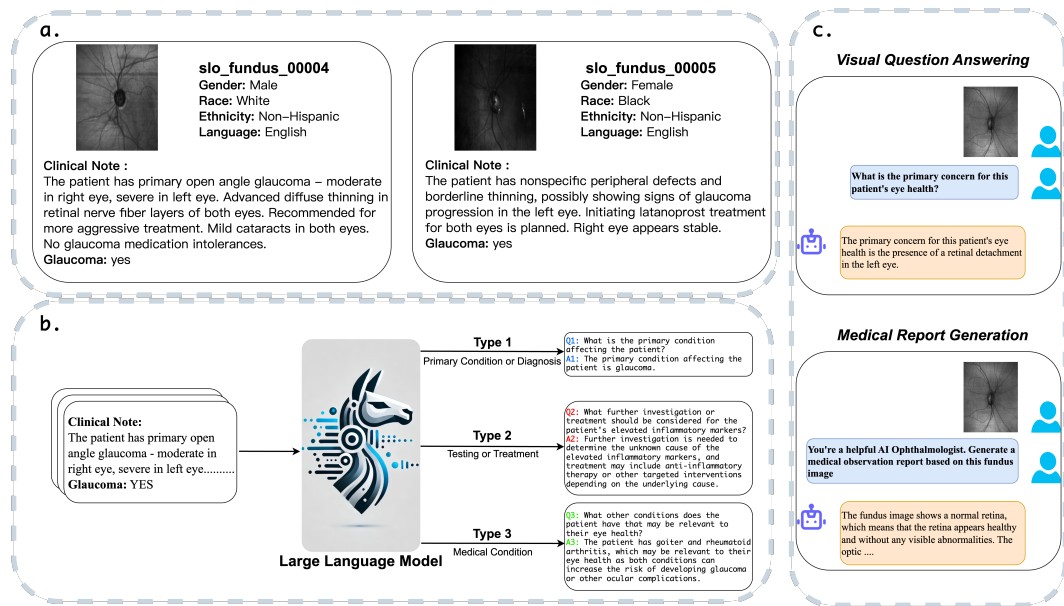

Figure 1: Overview of the FMBench QA pair construction. **(a)** This panel showcases two sample entries from the FMBench dataset, derived from the Harvard-FairVLMed dataset. Each entry features a fundus image paired with a clinical report and detailed demographic data. **(b)** Illustrated here is the LLM-based generation of QA pairs using Llama-3.1-70B-Instruct. The LLM queries clinical reports to produce QA pairs categorized into primary condition or diagnosis, testing or treatment, and medical condition. **(c)** The inference of QA pairs in VQA and the medical reports generation.

according to the sample size or other significance measures of each group. The weighted average performance ($\overline{\mathcal{G}}$) sets a baseline for measuring deviations among groups, incorporating a balance factor ($\lambda$), which moderates the trade-off between overall performance and fairness. Adjusting $\lambda$ allows for greater emphasis on fairness, albeit potentially impacting overall performance. Including the total number of demographic groups (N), FAP ensures that VQA systems are not only effective but also fair and inclusive, providing a comprehensive framework for evaluating AI systems across diverse populations.

$$\text{FAP} = \frac{\sum_{i=1}^{N} \mathcal{W}_i \cdot \mathcal{G}i}{\sum_{i=1}^{N} \mathcal{W}i} - \lambda \cdot \sqrt{\frac{\sum_{i=1}^{N} \mathcal{W}_i \cdot (\mathcal{G}i - \overline{\mathcal{G}})^2}{\sum_{i=1}^{N} \mathcal{W}_i}} \tag{1}$$

The second term in FAP quantifies the degree of inequality in performance between groups. When the performance of all groups ($\mathcal{G}_i$) closely matches the weighted average ($\overline{\mathcal{G}}$), this value approaches zero, indicating relatively even distribution of performance and, hence, greater fairness. Conversely, significant variations in $\mathcal{G}_i$ across groups suggest reduced fairness.

$$\sigma_w = \sqrt{\frac{\sum_{i=1}^{N} \mathcal{W}_i \cdot (\mathcal{G}i - \overline{\mathcal{G}})^2}{\sum_{i=1}^{N} \mathcal{W}_i}} \tag{2}$$

We utilize normalized results for comparing the second parameter because normalized weighted root mean square deviation facilitates fairer and more valid comparisons between different categories, unaffected directly by the magnitude of mean performance scores ($\overline{\mathcal{G}}$) for each category.

$$\delta_{\text{norm}} = \left(\frac{\sigma_w}{\overline{\mathcal{G}}}\right) \tag{3}$$

# 4 EXPERIMENTS CONFIGURATION

## 4.1 EVALUATED MODELS

We deploy all experiments on four NVIDIA A100 (80G) GPUs. For all MLLM generations, we set the temperature parameter to 0 to eliminate randomness during text generation. We evaluate a diverse set of MLLMs that are designed to handle various vision-language tasks, including VQA and report generation. The models in this study vary in parameter sizes, and task-specific capabilities.

**MiniCPM-V 2.6 (8B)** (Yao et al., 2024): This model integrates the SigLip-400M vision encoder with the Qwen2-7B text decoder LLM, comprising 400M parameters in the vision encoder and 7B in the text decoder, totaling 8B parameters. We utilize the 8B variant for our evaluations, which is pre-trained on a large-scale general visual-language dataset but has not been fine-tuned on medical data[2].

**InternVL2 (26B) & InternVL1.5 (26B)** (Chen et al., 2024): These models employ the InternViT-6B-448px-V1-5 vision encoder coupled with the internlm2-chat-20b LLM. We evaluate the 26B variant of each. InternVL2 is pre-trained on a web-scale multimodal dataset inclusive of medical data, whereas InternVL1.5 does not incorporate medical data during pre-training[3].

**LLaVA-Med (7B)** (Li et al., 2024): This model features the CLIP-ViT-L-336px vision encoder paired with the Mistral-7B-Instruct-v0.2 text decoder LLM, housing 7B parameters in the text decoder. We evaluate the 7B variant, which is specifically trained on biomedical data, including the PMC-VQA dataset[4].

**LLaVA1.6 (7B/13B) & LLaVA1.5 (7B/13B)** (Liu et al., 2024): These models integrate the CLIP-ViT-L-336px vision encoder with the Vicuna LLM text decoder, scaling to 13B parameters—4B in the vision encoder and 9B in the text decoder. We evaluate both the 7B and 13B variants. The models are pre-trained on large-scale multimodal datasets, although they are not specifically fine-tuned on medical data[5].

## 4.2 ZERO-SHOT EVALUATION

To assess the performance and fairness of MLLMs on VQA and report generation tasks, we conduct zero-shot evaluations across eight open-source MLLMs. For the VQA task, we utilize medical images and accompanying questions as instruction inputs to all MLLMs, comparing the generated text against the ground truth answers. For the report generation task, we employ the same textual instruction: `You are a professional Ophthalmologist. Please generate the clinical report for the given fundus image. `, using the medical image as input to the MLLMs to generate clinical reports. These are then compared with the clinical reports from the original sample. We evaluate their performance using lexical metrics and LLM-based metrics for each demographic attribute and also assess their fairness using our proposed Fairness-Aware Performance (FAP) metric. We all know that in clinical applications, the medical report generation task is more difficult and important compared to medical VQA. However, mainstream medical MLLM research currently focuses predominantly on medical VQA tasks. Through our FMBench, we hope to bring new thinking to the community and conduct in-depth research on such tasks, highlighting the crucial need for advancements in medical report generation.

## 4.3 EVALUATION METRICS

**Lexical Metrics.** We assess the VQA and report generation tasks using nine metrics: BLEU 1-4 (Papineni et al., 2002), METEOR (Banerjee & Lavie, 2005), ROUGE 1-2, ROUGE-L (Lin, 2004), and CIDEr-D. BLEU measures literal accuracy, METEOR accounts for both accuracy and fluency, ROUGE-L evaluates sentence structure and fluency, ROUGE-1 and ROUGE-2 assess uni-gram and bi-gram overlaps, and CIDEr-D evaluates the relevance and uniqueness of the generated content.

---

[2]https://huggingface.co/openbmb/MiniCPM-V-2_6

[3]https://github.com/OpenGVLab/InternVL

[4]https://huggingface.co/microsoft/llava-med-v1.5-mistral-7b

[5]https://huggingface.co/liuhaotian/llava-v1.6-vicuna-13b

```
Now you are a professional ophthalmologist!

Please refer to the ground truth and prediction based on
the following two paragraphs, identify the aspects
mentioned in the ground truth and calculate the
percentage of these aspects that are either correctly
mentioned or partially matched in the prediction, scoring
from 0 to 100.
The answer must be short and precise.

Ground Truth:{ground_truth} Prediction:{prediction}

The output format is:
The Score is "xx".
```

Figure 2: The prompt for LLM scoring. Lexical metrics fall short in evaluating the semantic correctness of VQA and report generation tasks. To overcome this limitation, we directly query an LLM to score the generated results, utilizing Llama-3.1-70B-Instruct (Meta, 2024) for this purpose.

However, these metrics primarily focus on word-level accuracy and lack sufficient consideration of context, factual correctness, and overall sentence semantics, which are crucial in medical tasks.

**GREEN Score.** Given that lexical metrics alone are insufficient for accurately evaluating the clinical relevance of generated text in medical tasks, we adopt the GREEN (Generative Radiology Evaluation and Error Notation) metric (Ostmeier et al., 2024). This metric is designed to simulate clinical expert evaluations by comparing generated text with reference text, focusing on factual accuracy and semantic coherence. It ranges from 0 to 1, with higher scores indicating greater semantic similarity and coherence between the generated and reference texts. The GREEN metric is implemented using an LLM[6].

**LLM Scoring.** To further assess the generated and reference texts, we utilize a powerful LLM following (Bai et al., 2024). As shown in Figure 2, we employ Llama-3.1-70B-Instruct to generate subjective scores ranging from 0 to 100, with higher scores reflecting better performance.

**Fairness-Aware Performance (FAP).** While various metrics are used to evaluate the correctness of generated and reference texts, they do not address the fairness of MLLMs across different demographic groups. To remedy this, we introduce the FAP metric, specifically designed to evaluate fairness.

## 5 RESULTS

In this section, we present and analyze the performance of eight MLLMs on two tasks: zero-shot Visual Question Answering (VQA) and zero-shot report generation. Additionally, we evaluate their fairness across four demographic attributes.

### 5.1 BENCHMARKING MLLM PERFORMANCE

**Zero-shot VQA.** We first investigate the performance of MLLMs on the zero-shot VQA task by averaging nine lexical metrics across all demographic groups, as shown in Figure 3 (top left). LLaVA-Med achieves the highest lexical score. However, as shown in Figure 4, assessing performance solely at the word level can lead to misinterpretations of outcomes. To address this limitation, it is essential to utilize LLM scores and GREEN scores, which evaluate results at the semantic level, thereby enhancing the accuracy of evaluations for MLLM outputs.

However, when evaluating with the GREEN and LLM scores (Figure 3, bottom left), MiniCPM-V-2.6 substantially outperforms LLaVA-Med, indicating a disparity between lexical and semantic performance. Moreover, larger-scale models fail to consistently demonstrate performance improvements.

As depicted in samples 4 to 6 of Figure 4, despite LLaVA-Med being trained on extensive medical datasets, it primarily acquires relevant terminology and words. However, it demonstrates limitations in its semantic understanding and generalization capabilities, struggling to effectively comprehend and respond to new medical queries. Therefore, it is crucial to ensure that MLLMs learn to understand medical problems, not just the relevant terminology and words.

**Zero-shot Report Generation** For the report generation task, as depicted in Figure 3 (top right), all MLLMs exhibit very poor lexical performance, including LLaVA-Med, which has been trained on

---

[6]https://huggingface.co/datasets/StanfordAIMI/GREEN

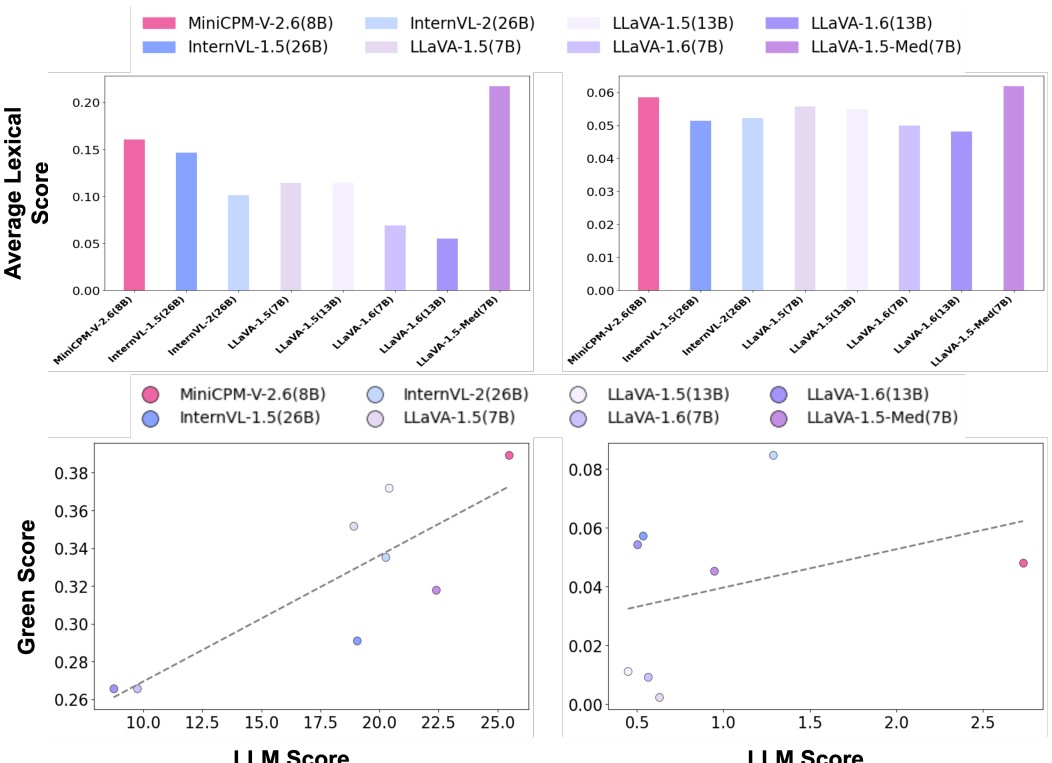

Figure 3: Performance of MLLMs averaged across all demographic attributes. The dashed line shows the relationship between the GREEN and LLM scores. **Top Left:** Average of 9 lexical scores and demographics on the zero-shot VQA task. **Top Right:** Average of 9 lexical scores and demographics on the zero-shot report generation task. **Bottom Left:** Correlation between GREEN and LLM scores on the zero-shot VQA task. **Bottom Right:** Correlation between GREEN and LLM scores on the zero-shot report generation task.

15 million medical data points. Similarly, the LLM-based metric (Figure 3, bottom right) reflects poor performance across the board, with no significant gains from larger models. This demonstrates the current MLLMs' incapability in the zero-shot report generation task.

In summary, current MLLMs perform poorly on both zero-shot VQA and report generation tasks, even those trained on substantial medical data. Surprisingly, general MLLMs such as MiniCPM and InternVL, despite not being specifically tuned for medical data, show competitive performance, even surpassing some medical-specific MLLMs. This suggests that a well-designed general MLLM can perform well on medical tasks without targeted training. Additionally, increasing model scale does not necessarily lead to performance gains, indicating that brute-force scaling is not an ideal solution for improving MLLM performance.

## 5.2 BENCHMARKING MLLM FAIRNESS

**Zero-shot VQA.** We evaluated the fairness of eight MLLMs on the zero-shot VQA task using the Fairness-Adjusted Performance (FAP) score, as depicted in Figure 5 (left). MiniCPM-V 2.6 demonstrates the best balance across different demographic attributes, consistently producing high-quality outputs and exhibiting superior fairness.

Furthermore, MiniCPM-V 2.6 achieves the highest scores across all attributes, considering the distribution of data across different groups. However, we observe higher deviations in the Race and Gender attributes, suggesting that even general MLLMs like MiniCPM-V 2.6 still struggle with maintaining fairness across all attributes in the VQA task. This indicates that biases inherent in the training data continue to impact the performance of MLLMs.

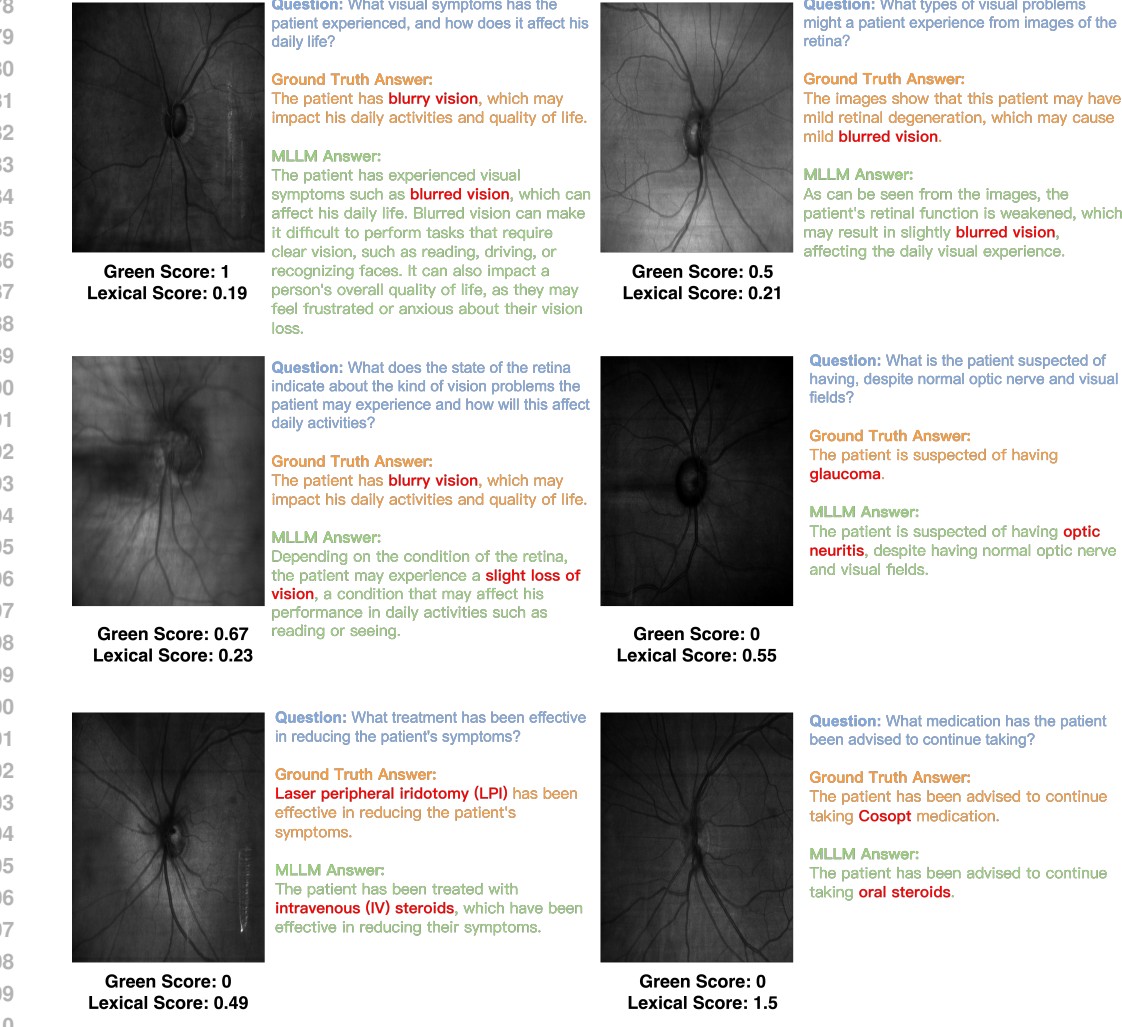

Figure 4: We provide four samples from LLaVA-Med inference results. **Sample 1-3:** We can see that the ground truth answers and the predicted answers are higly semantic consistent. **Sample 4-6:** Ground truth answers and predicted answers consistent at word-level but different in semantics.

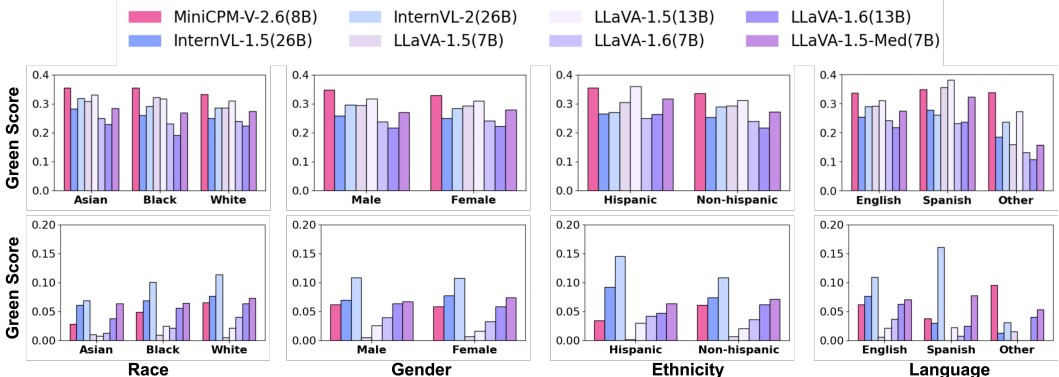

Figure 5: GREEN scores for 8 MLLMs across different demographic groups. **Top:** GREEN scores for the zero-shot VQA task. **Bottom:** GREEN scores for the zero-shot report generation task.

**Zero-shot Report Generation.** As depicted in Figure 5, all MLLMs exhibit poor performance on fairness in the report generation task, particularly concerning the language attribute. Further analy-

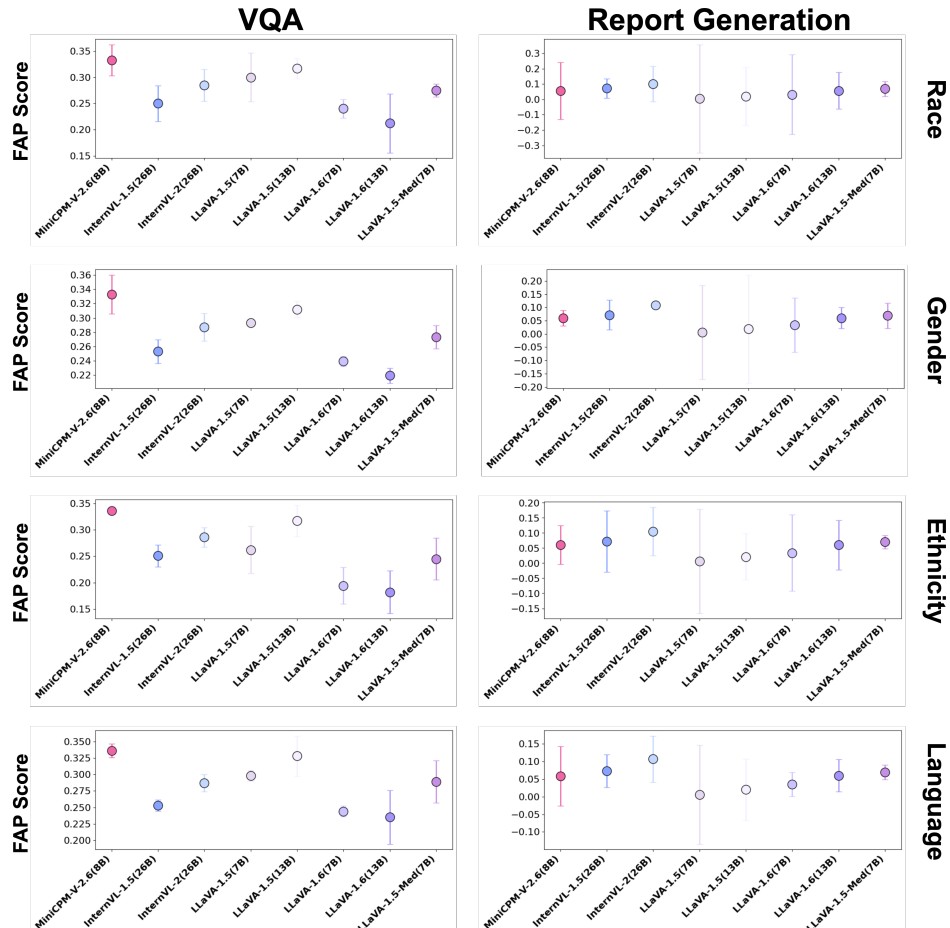

Figure 6: FAP scores for 8 MLLMs across four demographic attributes. **Left:** Performance on the zero-shot VQA task. **Right:** Performance on the zero-shot report generation task.

sis, shown in Figure 6, reveals that the Fairness-Adjusted Performance (FAP) scores are consistently low across all MLLMs, with significant deviations observed. Moreover, all MLLMs experience more pronounced fluctuations in performance during the report generation task compared to the VQA task. This is likely due to the complexity of creating detailed clinical reports as opposed to merely answering specific questions. These findings underscore that current MLLMs are inadequate at ensuring fairness in the report generation task.

## 6  CONCLUSION

In this work, we introduce FMBench, the first benchmark designed to evaluate the fairness of Multi-modal Large Language Models (MLLMs) on medical multimodal tasks. FMBench includes four demographic attributes, encompassing ten groups in total, and features 30,000 image-question-answer pairs for VQA evaluation and 10,000 image-report pairs for report generation. It provides a comprehensive assessment of MLLM fairness across these tasks. We also identify limitations in current metrics for fairly evaluating VQA and report generation and propose the Fairness-Adjusted Performance (FAP) score as a new metric for assessing fairness. Our findings indicate that existing MLLMs demonstrate unstable performance across demographic groups, even when trained on large-scale, diverse datasets. Moreover, their performance on both VQA and report generation tasks is unsatisfactory. Notably, we observe that medical-specific MLLMs generate text with high lexical accuracy but low semantic correctness (as indicated by the GREEN score), while general MLLMs like MiniCPM produce more semantically accurate text but with lower lexical scores. This discrepancy reveals the shortcomings of current metrics and underscores that current medical MLLMs

often mimic medical style without truly understanding medical content. We hope that FMBench and the FAP score will assist the research community in better evaluation practices and encourage the development of fairer and more capable MLLMs.

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

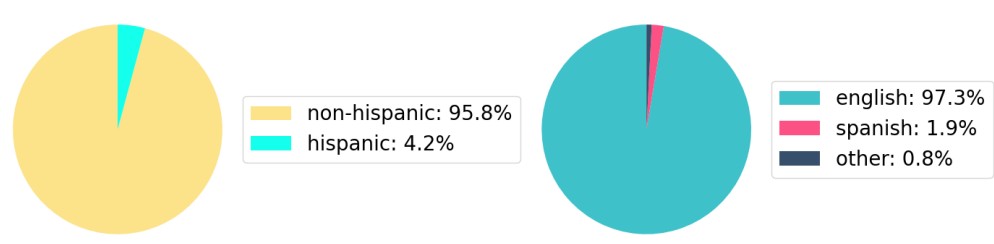

Figure 7: Data distribution of different demographic attributes **(a)** Gender. **(b)** Race. **(c)** Ethnicity. **(c)** Language.

## A  DATASET DETAILS

We utilize open-source medical visual language datasets to construct the FMBench benchmarks, which encompass two critical tasks: medical Visual Question Answering (VQA) and medical report generation. Specifically, our dataset incorporates four different demographic attributes: gender, ethnicity, race, and language, as illustrated in Figure 7. We employ these open-source datasets to establish a comprehensive benchmark under the FMBench framework.

We generated a total of 30,000 QA pairs and 10,000 image-report pairs, featuring textual high-frequency words, as illustrated in Figure 8. This data was utilized to generate the medical Visual Question Answering (VQA) tasks and medical reports.

## B  IMPLEMENTATION DETAILS

We conducted the experiments on four NVIDIA A100 (80G) GPUs. We benchmarked eight open-source Multimodal Large Language Models (MLLMs) with default settings, including MiniCPM-V 2.6 (8B) (Yao et al., 2024), InternVL2 (26B), InternVL1.5 (26B) (Chen et al., 2024), LLaVA-Med (7B) (Li et al., 2024), LLaVA1.6 (7B), LLaVA1.6 (13B), LLaVA1.5 (7B), and LLaVA1.5

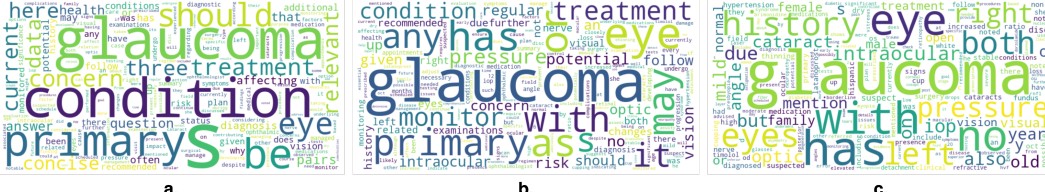

Figure 8: Word cloud of the FMBench datasets. **(a)** Question of the Visual Question Answer. **(b)** Answer of the Visual Question Answer. **(c)** Clinical note of the Medical Report Generation.

Table 2: Zero-shot Lexical Score of VQA.

| Models | Params | BLEU-1 | BLEU-2 | BLEU-3 | BLEU-4 | METEOR | ROUGE-L | ROUGE-1 | ROUGE-2 | CIDEr-D |
|---|---|---|---|---|---|---|---|---|---|---|
| MiniCPM-V-2.6 | 8B | 0.210 | 0.077 | 0.043 | 0.027 | 0.346 | 0.211 | 0.270 | 0.113 | 0.149 |
| InternVL-V2.0 | 26B | 0.177 | 0.073 | 0.042 | 0.028 | 0.313 | 0.190 | 0.229 | 0.109 | 0.159 |
| InternVL-Chat-V1-5 | 26B | 0.125 | 0.041 | 0.021 | 0.012 | 0.283 | 0.149 | 0.186 | 0.076 | 0.017 |
| llava-v1.5 | 7B | 0.135 | 0.052 | 0.028 | 0.017 | 0.295 | 0.160 | 0.196 | 0.085 | 0.062 |
| llava-v1.5 | 13B | 0.135 | 0.052 | 0.028 | 0.017 | 0.299 | 0.161 | 0.198 | 0.087 | 0.053 |
| llava-v1.6 | 7B | 0.074 | 0.025 | 0.013 | 0.008 | 0.214 | 0.094 | 0.120 | 0.045 | 0.028 |
| llava-v1.6 | 13B | 0.057 | 0.017 | 0.007 | 0.004 | 0.198 | 0.077 | 0.100 | 0.034 | 0.003 |
| llava-med-v1.5 | 7B | **0.277** | **0.123** | **0.077** | **0.051** | **0.360** | **0.268** | **0.314** | **0.159** | **0.326** |

Table 3: Lexical Metrics of Zero-shot Medical Report Generation . If the value lower than 0.001 we will not consider it is an valid data and using '-' to present it.

| Models | Size | BLEU-1 | BLEU-2 | BLEU-3 | BLEU-4 | METEOR | ROUGE-L | ROUGE-1 | ROUGE-2 | CIDEr-D |
|---|---|---|---|---|---|---|---|---|---|---|
| MiniCPM-V-2.6 | 8B | 0.134 | 0.007 | - | - | **0.169** | 0.085 | 0.127 | 0.004 | 0.001 |
| InternVL-V2.0 | 26B | 0.119 | 0.007 | - | - | 0.149 | 0.077 | 0.105 | 0.005 | 0.001 |
| InternVL-Chat-V1-5 | 26B | 0.113 | 0.007 | - | - | 0.155 | 0.079 | 0.109 | 0.006 | 0.001 |
| llava-v1.5 | 7B | 0.136 | 0.008 | - | - | 0.148 | 0.085 | 0.116 | 0.007 | 0.002 |
| llava-v1.5 | 13B | 0.124 | 0.008 | - | - | 0.154 | 0.084 | 0.114 | 0.009 | 0.001 |
| llava-v1.6 | 7B | 0.110 | 0.006 | - | - | 0.147 | 0.077 | 0.102 | 0.007 | 0.001 |
| llava-v1.6 | 13B | 0.100 | 0.006 | - | - | 0.149 | 0.072 | 0.098 | 0.007 | 0.001 |
| llava-med-v1.5 | 7B | **0.154** | **0.010** | **0.001** | - | 0.152 | **0.093** | **0.131** | **0.010** | **0.006** |

(13B) (Liu et al., 2024). All model checkpoints can be downloaded from Hugging Face: https://huggingface.co/. Specific download links are provided below:

- MiniCPM-V 2.6 (8B): https://huggingface.co/openbmb/MiniCPM-V-2_6
- InternVL2 (26B): https://huggingface.co/OpenGVLab/InternVL2-26B
- InternVL1.5 (26B): https://huggingface.co/OpenGVLab/InternVL-Chat-V1-5
- LLaVA-Med (7B): https://huggingface.co/microsoft/llava-med-v1.5-mistral-7b
- LLaVA1.6 (7B): https://huggingface.co/liuhaotian/llava-v1.6-vicuna-7b
- LLaVA1.6 (13B): https://huggingface.co/liuhaotian/llava-v1.6-vicuna-13b
- LLaVA1.5 (7B): https://huggingface.co/liuhaotian/llava-v1.5-7b
- LLaVA1.5 (13B): https://huggingface.co/liuhaotian/llava-v1.5-13b

## C  MORE RESULTS OF ZERO-SHOT EVALUATION

### C.1  LEXICAL RESULTS DETAILS

In this section, we present the details of nine lexical metrics used to evaluate the performance of Multimodal Large Language Models (MLLMs). Figure 3 provides a visual representation of these metrics. Additionally, the results for each metric are systematically tabulated in Table 2 and Table 3.

## C.2 GREEN SCORE RESULTS DETAILS

In this section, we detail the GREEN scores for each demographic group as depicted in Figure 5. The results are further analyzed in Table 4 and Table 5, providing an in-depth examination of performance across four demographic attributes using eight open-source Multimodal Large Language Models (MLLMs).

## C.3 FAP SCORE RESULTS DETAILS

In this section, we present detailed results for the Fairness-Aware Performance (FAP) and Normalized Deviation, as illustrated in Figure 6. The analysis of these metrics, based on four demographic attributes across eight open-source Multimodal Large Language Models (MLLMs), is systematically detailed in Table 6 and Table 7.

Table 4: GREEN scores on Zero-shot VQA task with different demographic attributes.

| Attribute | Model | Average Metric |
|---|---|---|
| Race | MiniCPM-V-2_6 | Asian: 0.356, Black: 0.355, White: 0.332 |
| | InternVL-Chat-V1-5 | Asian: 0.283, Black: 0.260, White: 0.251 |
| | InternVL2-26B | Asian: 0.319, Black: 0.291, White: 0.286 |
| | llava-1.5-7b | Asian: 0.309, Black: 0.322, White: 0.286 |
| | llava-1.5-13b | Asian: 0.331, Black: 0.318, White: 0.310 |
| | llava-1.6-7b | Asian: 0.250, Black: 0.232, White: 0.240 |
| | llava-1.6-13b | Asian: 0.229, Black: 0.191, White: 0.225 |
| | llava-1.5-med-7b | Asian: 0.284, Black: 0.269, White: 0.275 |
| Gender | MiniCPM-V-2_6 | Male: 0.348, Female: 0.329 |
| | InternVL-Chat-V1-5 | Male: 0.259, Female: 0.251 |
| | InternVL2-26B | Male: 0.296, Female: 0.284 |
| | llava-1.5-7b | Male: 0.295, Female: 0.293 |
| | llava-1.5-13b | Male: 0.317, Female: 0.310 |
| | llava-1.6-7b | Male: 0.238, Female: 0.241 |
| | llava-1.6-13b | Male: 0.217, Female: 0.222 |
| | llava-1.5-med-7b | Male: 0.270, Female: 0.279 |
| Ethnicity | MiniCPM-V-2_6 | Hispanic: 0.355, Non-hispanic: 0.337 |
| | InternVL-Chat-V1-5 | Hispanic: 0.265, Non-hispanic: 0.254 |
| | InternVL2-26B | Hispanic: 0.271, Non-hispanic: 0.290 |
| | llava-1.5-7b | Hispanic: 0.305, Non-hispanic: 0.294 |
| | llava-1.5-13b | Hispanic: 0.360, Non-hispanic: 0.312 |
| | llava-1.6-7b | Hispanic: 0.250, Non-hispanic: 0.240 |
| | llava-1.6-13b | Hispanic: 0.264, Non-hispanic: 0.218 |
| | llava-1.5-med-7b | Hispanic: 0.318, Non-hispanic: 0.273 |
| Language | MiniCPM-V-2_6 | English: 0.337, Spanish: 0.349, Other: 0.338 |
| | InternVL-Chat-V1-5 | English: 0.254, Spanish: 0.277, Other: 0.184 |
| | InternVL2-26B | English: 0.290, Spanish: 0.261, Other: 0.237 |
| | llava-1.5-7b | English: 0.291, Spanish: 0.356, Other: 0.159 |
| | llava-1.5-13b | English: 0.310, Spanish: 0.381, Other: 0.273 |
| | llava-1.6-7b | English: 0.241, Spanish: 0.232, Other: 0.131 |
| | llava-1.6-13b | English: 0.218, Spanish: 0.237, Other: 0.107 |
| | llava-1.5-med-7b | English: 0.274, Spanish: 0.323, Other: 0.158 |

Table 5: GREEN scores of Zero-shot Medical Report Generation task with different demographic attributes.

| Attribute | Model | Average Metric |
|---|---|---|
| Race | MiniCPM-V-2_6 | Asian: 0.029, Black: 0.049, White: 0.066 |
| | InternVL-Chat-V1-5 | Asian: 0.061, Black: 0.069, White: 0.077 |
| | InternVL2-26B | Asian: 0.069, Black: 0.101, White: 0.114 |
| | llava-1.5-7b | Asian: 0.011, Black: 0.010, White: 0.005 |
| | llava-1.5-13b | Asian: 0.008, Black: 0.025, White: 0.022 |
| | llava-1.6-7b | Asian: 0.013, Black: 0.022, White: 0.041 |
| | llava-1.6-13b | Asian: 0.038, Black: 0.056, White: 0.064 |
| | llava-1.5-med-7b | Asian: 0.064, Black: 0.065, White: 0.073 |
| Gender | MiniCPM-V-2_6 | Male: 0.062, Female: 0.059 |
| | InternVL-Chat-V1-5 | Male: 0.070, Female: 0.078 |
| | InternVL2-26B | Male: 0.109, Female: 0.108 |
| | llava-1.5-7b | Male: 0.005, Female: 0.007 |
| | llava-1.5-13b | Male: 0.026, Female: 0.017 |
| | llava-1.6-7b | Male: 0.040, Female: 0.033 |
| | llava-1.6-13b | Male: 0.064, Female: 0.059 |
| | llava-1.5-med-7b | Male: 0.067, Female: 0.074 |
| Ethnicity | MiniCPM-V-2_6 | Hispanic: 0.035, Non-hispanic: 0.061 |
| | InternVL-Chat-V1-5 | Hispanic: 0.092, Non-hispanic: 0.074 |
| | InternVL2-26B | Hispanic: 0.146, Non-hispanic: 0.109 |
| | llava-1.5-7b | Hispanic: 0.002, Non-hispanic: 0.007 |
| | llava-1.5-13b | Hispanic: 0.030, Non-hispanic: 0.021 |
| | llava-1.6-7b | Hispanic: 0.042, Non-hispanic: 0.036 |
| | llava-1.6-13b | Hispanic: 0.048, Non-hispanic: 0.062 |
| | llava-1.5-med-7b | Hispanic: 0.064, Non-hispanic: 0.072 |
| Language | MiniCPM-V-2_6 | English: 0.062, Spanish: 0.038, Other: 0.096 |
| | InternVL-Chat-V1-5 | English: 0.077, Spanish: 0.030, Other: 0.013 |
| | InternVL2-26B | English: 0.110, Spanish: 0.161, Other: 0.031 |
| | llava-1.5-7b | English: 0.006, Spanish: 0.0, Other: 0.016 |
| | llava-1.5-13b | English: 0.022, Spanish: 0.023, Other: 0.0 |
| | llava-1.6-7b | English: 0.037, Spanish: 0.008, Other: 0.0 |
| | llava-1.6-13b | English: 0.063, Spanish: 0.025, Other: 0.041 |
| | llava-1.5-med-7b | English: 0.071, Spanish: 0.078, Other: 0.054 |

Table 6: FAP Values and Normalized Deviations in Zero-shot VQA

| Model | Category | FAP Value | Normalized Deviation(%) |
|---|---|---|---|
| MiniCPM-V-2_6 | Race | 0.333 | 2.968 |
| | Gender | 0.333 | 2.750 |
| | Language | 0.336 | 0.443 |
| | Ethnicity | 0.336 | 1.031 |
| InternVL-Chat-V1-5 | Race | 0.250 | 3.427 |
| | Gender | 0.253 | 1.670 |
| | Language | 0.251 | 2.088 |
| | Ethnicity | 0.253 | 0.862 |
| InternVL2-26B | Race | 0.285 | 3.059 |
| | Gender | 0.287 | 1.928 |
| | Language | 0.286 | 1.870 |
| | Ethnicity | 0.287 | 1.327 |
| llava-1.5-7b | Race | 0.300 | 4.676 |
| | Gender | 0.293 | 0.365 |
| | Language | 0.262 | 4.470 |
| | Ethnicity | 0.298 | 0.777 |
| llava-1.5-13b | Race | 0.317 | 1.936 |
| | Gender | 0.312 | 1.127 |
| | Language | 0.317 | 3.017 |
| | Ethnicity | 0.328 | 3.064 |
| llava-1.6-7b | Race | 0.240 | 1.742 |
| | Gender | 0.239 | 0.681 |
| | Language | 0.194 | 3.473 |
| | Ethnicity | 0.244 | 0.802 |
| llava-1.6-13b | Race | 0.212 | 5.619 |
| | Gender | 0.219 | 1.051 |
| | Language | 0.182 | 4.014 |
| | Ethnicity | 0.235 | 4.073 |
| llava-1.5-med-7b | Race | 0.275 | 1.259 |
| | Gender | 0.273 | 1.613 |
| | Language | 0.245 | 3.944 |
| | Ethnicity | 0.289 | 3.177 |

Table 7: FAP Values and Normalized Deviations on Zero-shot Medical Report Generation Task

| Model | Category | FAP Value | Normalized Deviation(%) |
|---|---|---|---|
| MiniCPM-V-2_6 | Race | 0.055 | 18.595 |
| | Gender | 0.060 | 2.885 |
| | Language | 0.060 | 6.404 |
| | Ethnicity | 0.058 | 8.443 |
| InternVL-Chat-V1-5 | Race | 0.072 | 6.325 |
| | Gender | 0.072 | 5.560 |
| | Language | 0.072 | 10.111 |
| | Ethnicity | 0.073 | 4.662 |
| InternVL2-26B | Race | 0.102 | 11.518 |
| | Gender | 0.108 | 0.527 |
| | Language | 0.105 | 8.014 |
| | Ethnicity | 0.107 | 6.592 |
| llava-1.5-7b | Race | 0.005 | 35.312 |
| | Gender | 0.006 | 17.787 |
| | Language | 0.006 | 17.230 |
| | Ethnicity | 0.006 | 14.061 |
| llava-1.5-13b | Race | 0.019 | 18.802 |
| | Gender | 0.019 | 20.454 |
| | Language | 0.021 | 7.634 |
| | Ethnicity | 0.020 | 8.781 |
| llava-1.6-7b | Race | 0.031 | 26.043 |
| | Gender | 0.034 | 10.325 |
| | Language | 0.034 | 12.563 |
| | Ethnicity | 0.035 | 3.448 |
| llava-1.6-13b | Race | 0.057 | 12.051 |
| | Gender | 0.060 | 3.986 |
| | Language | 0.060 | 8.214 |
| | Ethnicity | 0.060 | 4.588 |
| llava-1.5-med-7b | Race | 0.069 | 4.949 |
| | Gender | 0.069 | 4.769 |
| | Language | 0.070 | 2.221 |
| | Ethnicity | 0.070 | 2.073 |

