# OpenReview forum: "FMBench: Benchmarking Fairness in Multimodal Large Language Models on Medical Tasks"
_ICLR.cc/2025/Conference — ICLR 2025 Conference Withdrawn Submission_

### Official Review · Reviewer_2vad · 2024-11-01

**Soundness:** 2
**Presentation:** 1
**Contribution:** 2
**Rating:** 1
**Confidence:** 5

**Summary:**

This paper introduces FMBench, the first benchmark specifically designed to evaluate the fairness of multimodal large language models (MLLM) on medical tasks. FMBench aims to assess the fairness of these models across different demographic characteristics (including race, gender, language, and ethnicity), which is crucial for clinical applications as bias can lead to unequal medical outcomes. It encompasses four demographic characteristics: race, gender, language, and ethnicity, and evaluates fairness across two tasks - visual question answering (VQA) and report generation (RG). The VQA task adopts an open-ended question-answering format to enhance practical applicability and reduce bias. It introduces "Fairness-Aware Performance (FAP)" as a new metric to quantify MLLM fairness across different groups.

**Strengths:**

1. The experimental evaluation of 8 open-source MLLMs reveals significant performance variations across different demographic attributes, with particularly high inconsistency in language attributes. Additionally, traditional lexical evaluation methods (such as BLEU, METEOR) are insufficient for open-ended multimodal tasks, necessitating supplementary semantic-based metrics like GREEN scores and LLM scoring.

2.The paper reveals performance disparities of existing MLLMs across different demographic groups, finds that increasing model scale does not necessarily lead to performance improvements, and identifies key challenges in medical report generation tasks.

**Weaknesses:**

1. Lack of few-shot and fine-tuning experiments.

2. Both GREEN score and LLM score rely on the LLaMA-3.1-70B-Instruct model. Using a single model as the evaluation standard may introduce bias, so it is recommended to use multiple different LLM models for cross-validation.

3.The research lacks depth; different types of medical imaging (e.g., X-ray, CT, MRI) may present different fairness issues. Relevant literature on MLLM in this regard should be supplemented for discussion.

You can refer to recent literature on some prompt experiments, titled "MCPL: Multi-modal Collaborative Prompt Learning for Medical Vision-Language Model," (https://ieeexplore.ieee.org/document/10570257) which provides a more detailed experimental procedure.

**Questions:**

This paper doesn't present particularly innovative research. Although it compares the fairness issues of multimodal question answering models, the study would be more complete if it compared different datasets for analyzing these issues.

---

### Official Review · Reviewer_u2tc · 2024-11-02

**Soundness:** 2
**Presentation:** 2
**Contribution:** 2
**Rating:** 5
**Confidence:** 3

**Summary:**

The paper introduces FMBench, a new benchmark designed to evaluate the fairness and performance of Multimodal Large Language Models (MLLMs) in medical visual question answering (VQA) and report generation (RG) tasks. Built from the Harvard-FairVLMed dataset, which includes 10,000 fundus images paired with clinical reports and demographic data, FMBench enables performance assessments across race, gender, ethnicity, and language groups. The authors detail their process for generating QA pairs using Llama-3.1-Instruct-70B and introduce the Fairness-Aware Performance (FAP) metric to measure group performance disparities.

**Strengths:**

- The work effectively addresses the critical gap in fairness evaluation of MLLMs, especially within the biomedical domain, where biases can have significant consequences. By facilitating fairness assessments across various demographic groups, FMBench promotes the development of more inclusive and equitable AI systems.

- The paper introduces an innovative benchmark for assessing fairness in MLLMs, focusing on important demographic categories such as race, gender, ethnicity, and language. This comprehensive approach sets a new standard for fairness evaluation in multimodal AI.

- The authors propose a new metric, FAP, which accounts for the performance of each group as well as the relative performance compared to the weighted average across all groups. This metric helps identify disparities between demographic groups relative to the overall mean.

- The paper includes a good number of baseline comparisons, incorporating both biomedical and general-domain benchmarks. Evaluating the fairness of general-domain MLLMs is also crucial, as these models are frequently repurposed for biomedical applications.

**Weaknesses:**

- The paper lacks sufficient evaluation of biomedical multimodal models. Although the benchmark is designed for the biomedical domain, only one model (Llava-Med) trained on biomedical data is evaluated. The authors should consider including other biomedical models, such as Dragonfly-Med ([link](https://huggingface.co/togethercomputer/Llama-3.1-8B-Dragonfly-Med-v2)), to provide a more comprehensive evaluation.

- Including the performance of commercial MLLMs such as Claude and GPT could enhance the study. These models are widely used by the general public, including for biomedical inquiries. Adding their performance would strengthen the paper by offering insights into the fairness of commercial MLLMs and their application to medical tasks.

- The paper does not clearly explain how to choose the value of lambda in the FAP equation, nor does it specify which value was used for their model evaluation. Providing more details or guidelines for selecting lambda would improve clarity and reproducibility.

- The paper makes bold claims about the performance of general MLLMs based on limited evaluations. For instance, in section 5.1, the authors state: *"Surprisingly, general MLLMs such as MiniCPM and InternVL, despite not being specifically tuned for medical data, show competitive performance, even surpassing some medical-specific MLLMs. This suggests that a well-designed general MLLM can perform well on medical tasks without targeted training. Additionally, increasing model scale does not necessarily lead to performance gains, indicating that brute-force scaling is not an ideal solution for improving MLLM performance."* These claims are significant yet are drawn from evaluations on only one type of biomedical image task and a single biomedical model. Expanding the range of tasks and models evaluated would lend stronger support to these assertions.

**Questions:**

- How did the authors control for potential hallucinations from the LLM while generating their dataset? Was manual inspection of a handful of examples done?
- What lambda value did the authors use while calculating their FAP metric? Could they also show the plot with only the second term in their FAP equation, which measures the deviation of performance of each group from the weighted mean?
- Why did the authors only consider one biomedical model in their evaluation, even though their benchmark is specifically tailored towards multimodal biomedical models?
- Have the authors considered evaluating the performance of commercial MLLMs, such as Claude and GPT, to provide insights into how these widely used models perform in terms of fairness?

---

### Official Review · Reviewer_V8yT · 2024-11-02

**Soundness:** 3
**Presentation:** 2
**Contribution:** 2
**Rating:** 5
**Confidence:** 5

**Summary:**

This work introduces FMBench, a benchmark to evaluate fairness in medical multimodal tasks across demographic attributes. It also proposes the Fairness-Aware Performance (FAP) metric to assess equitable performance of medical language models (MLLMs) for open-form tasks. Eight MLLMs are evaluated, revealing traditional metrics’ limitations and inconsistencies with clinician-verified assessments. Results show that all models demonstrate varying performance across demographics, indicating potential fairness risks.

**Strengths:**

-The authors construct a benchmark in the opthamoogy domain coupled with demographic attributes to address to fairness issues in medical AI applications.
- The authors assess two tasks in their benchmark (generation and VQA)
-The study reveals limitations of traditional metrics, showing they can be inconsistent with clinician-verified assessments.

**Weaknesses:**

- Only one dataset is used (Harvard-FairVLMed dataset) to perform the benchmark. Also a single domain opthamalogy is being present.
- Paper should be reframed as it mentions its the first to assess fairness in medical multimodal tasks across demographic groups. However, it only uses fundus images.
- There are datasets that are paired with demographic attributes e.g language, race, age, etc . that could be utilized to ensure robustness. For example, Medical Information Mart for Intensive Care (MIMIC) dataset which could be used.
- There have been previous works utilizing demographics to perform fairness evaluation on multimodal datasets. The related works could benefit from discussing datasets and multimodality which use model bias across different demographic subgroups, including race-ethnicity, sex, and age: https://www.thelancet.com/journals/landig/article/PIIS2589-7500(22)00063-2/fulltext?ref=9gag
Further, to assess fairness other works have presented various dimensions of fairness going beyond of demographic groups, such as: https://arxiv.org/pdf/2403.18196
- Releasing the dataset would contribute to medical community, but not clear if it's clinically verified and it only consists of 4 attributes which limits the assessment.

**Questions:**

- Minor errors: Line 147 4 different demogrphy attributes.
- Where the reports clinically-verified for the assessment.

---

### Official Review · Reviewer_QV82 · 2024-11-07

**Soundness:** 3
**Presentation:** 3
**Contribution:** 1
**Rating:** 3
**Confidence:** 4

**Summary:**

This paper introduces FMBench, a benchmark designed to assess the fairness of multimodal large language models (MLLMs) in medical tasks like Visual Question Answering and Report Generation across diverse demographic groups. FMBench includes attributes such as race, ethnicity, language, and gender, and uses both lexical and clinical metrics, along with a new Fairness-Aware Performance (FAP) metric, to evaluate model fairness. The authors test eight MLLMs with FMBench, aiming to improve fair model evaluation in healthcare applications.

**Strengths:**

+ The study targets a timely and important area in addressing fairness concerns in Med (M)LLMs.
+ Presenting a new benchmarking dataset for evaluating fairness is appreciated.
+ The reported messages upon the empirical analysis are interesting, especially observing that medical-specific MLLMs generate text with high lexical accuracy but low semantic correctness and the opposite observation about general MLLMs.

**Weaknesses:**

+ The biggest concern about this study is its significance. In its current form, it's rather unclear how the study goes well beyond the source FairClip dataset.
+ The title and abstract claims target the medical domain in general, but the dataset seems to be only applicable to one domain. i.e., ophthalmology.
+ It is a bit unusual for such a study not to give reviewers access to the dataset. The authors promise to release everything upon acceptance.
+ Another major concern about this study is about solely relying on LLMs for generating the QAs and summaries. This approach is indeed scalable but concerns about hallucination and the reliability of such an approach, especially for a high-stakes domain like medicine, seem unaddressed.
+ The proposed new fairness measure (FAP) seems logical but not significantly different from the common parity-based measures of fairness.
+ The study sets the temperature to zero for all experiments. This seems to cause no variation to be reported in the results. It seems unclear whether solely relying on mean stat with no variation is reliable.

**Questions:**

+ Did the study use various prompting methods, such as chain of thoughts, in the experiments?

---

### Note · Authors · 2024-11-12

I have read and agree with the venue's withdrawal policy on behalf of myself and my co-authors.